# Delirium and its association with short-term outcomes in younger and older patients with acute heart failure

Jin H. Han[1,2]*, Candace D. McNaughton[3], William B. Stubblefield[1], Peter S. Pang[4], Phillip D. Levy[5], Karen F. Miller[1], Sarah Meram[4], Mette Lind Cole[4], Cathy A. Jenkins[6], Hadassah H. Paz[1], Kelly M. Moser[1], Alan B. Storrow[1], Sean P. Collins[1,2], for the Emergency Medicine Research and Outcomes Consortium Investigators¶

1 Center for Emergency Research and Innovation, Vanderbilt University Medical Center, Nashville, TN, United States of America, 2 Geriatric Research, Education, and Clinical Center (GRECC), Tennessee Valley Healthcare System, Nashville, TN, United States of America, 3 Department of Medicine, Sunnybrook Research Institute, ICES, University of Toronto, Toronto, ON, Canada, 4 Department of Emergency Medicine, Indiana University School of Medicine, Indianapolis, IN, United States of America, 5 Department of Emergency Medicine, Wayne State University School of Medicine, Detroit, MI, United States of America, 6 Department of Biostatistics, Vanderbilt University Medical Center, Nashville, TN, United States of America

¶ Membership of the author group can be found in the Acknowledgments.
* jin.h.han@vumc.org

**Data Availability Statement:** All relevant data are within the paper and its Supporting Information files.

## Abstract

Younger patients (18 to 65 years old) are often excluded from delirium outcome studies. We sought to determine if delirium was associated with short-term adverse outcomes in a diverse cohort of younger and older patients with acute heart failure (AHF). We conducted a multi-center prospective cohort study that included adult emergency department patients with confirmed AHF. Delirium was ascertained using the Brief Confusion Assessment Method (bCAM). The primary outcome was a composite outcome of 30-day all-cause death, 30-day all-cause rehospitalization, and prolonged index hospital length of stay. Multivariable logistic regression was performed, adjusting for demographics, cognitive impairment without delirium, and HF risk factors. Older age ($\geq$ 65 years old)*delirium interaction was also incorporated into the model. Odds ratios (OR) with their 95% confidence intervals (95%CI) were reported. A total of 1044 patients with AHF were enrolled; 617 AHF patients were < 65 years old and 427 AHF patients were $\geq$ 65 years old, and 47 (7.6%) and 40 (9.4%) patients were delirious at enrollment, respectively. Delirium was significantly associated with the composite outcome (adjusted OR = 1.64, 95%CI: 1.02 to 2.64). The older age*delirium interaction p-value was 0.47. In conclusion, delirium was common in both younger and older patients with AHF and was associated with poorer short-term outcomes in both cohorts. Younger patients with acute heart failure should be included in future delirium outcome studies.

## Introduction

Cognitive impairment occurs in 23% to 67% of older adults admitted to the hospital with acute heart failure (AHF) [1–3] and is associated with increased risk of mortality and readmission

**Funding:** This project was partially supported by the NCATS/NIH under award number UL1 TR000445. Drs. Han and Collins receive funding from the Geriatric, Research, Education, and Clinical Center (GRECC). The funders had no role in study design, data collection, and analysis, decision to publish, or preparation of the manuscript.

**Competing interests:** Jin H. Han receives research support from Bristol Myers Squibb, Boehringer Ingelheim, and Merck. Candace D. McNaughton receives research support from Pfizer. Phillip D. Levy is a consultant for Apex Innovations, AstraZeneca, Bristol Myers Squibb, Mespere, Novartis, Cardionomics, Baim Institute, Ortho Clinical Diagnostics, Roche Diagnostics, Siemens, and Hospital Quality Foundation. Dr. Levy also receives research support from Amgen, Arterez, Bristol Myers Squibb, Beckman Coulter, and the Blue Cross Blue Shield of Michigan Foundation (BCBSMF). Peter S. Pang is a consultant for Baxter, Bristol Myers Squibb, Heart Initiative (DSMB). He also receives research support from Bristol Myers Squibb, Roche, Novartis, OrthoDiagnostics, Beckman Coulter, and Abbott. Mette L. Cole is employed by Roche. Sean P. Collins is a consultant for Ortho Clinical Diagnostics, Boehringer Ingelheim, Roche, and Bristol Myers Squibb. He receives research support from Ortho Clinical Diagnostics, Bristol Myers Squibb, Novartis, Astra Zeneca. This does not alter our adherence to PLOS ONE policies on sharing data and materials.

[3]. However, whether cognitive impairment is acute or chronic has important clinical implications. The most common form of chronic cognitive impairment is Alzheimer's Disease and Related Dementias (ADRD), which is characterized by a gradual loss of cognition over years, is not precipitated by an underlying medical illness, and is not considered a medical emergency. Delirium, however, is the most common form of acute cognitive impairment and is characterized by an acute loss of cognition over hours to days. It is usually precipitated by an underlying medical illness and is a potential medical emergency [4].

Delirium affects 10 to 17% of older emergency department (ED) patients [5, 6] and 25% of older hospitalized patients [7]. In acutely-ill older adults, delirium is an independent predictor of accelerated cognitive and functional decline [8], higher death rates [6], and prolonged hospital length of stays [9]. Delirium's impact on long-term outcomes, however, may depend on its underlying etiology. Cirbus et al. reported that delirium secondary to metabolic and organ dysfunction was significantly associated with poorer six-month function in a broad cohort of ED patients [10]. Therefore, delirium's impact on the health trajectories of patients with AHF is unclear.

Importantly, most delirium studies exclude adults who are less than 65 years old [5, 6]. The average age of previously published AHF and delirium outcome studies ranged from 75 to 83 years old [11–15]. Younger patients with AHF may similarly be susceptible to developing delirium and adverse outcomes. These studies also excluded patients who were discharged from the ED and enrolled homogeneous cohorts that were predominantly White or Asian [11–15]. Therefore, we performed a multi-center prospective cohort study in an ethnically diverse cohort of young and old adult ED patients. We sought to determine the frequency in which delirium and cognitive impairment without delirium occurred at enrollment in younger and older patients with AHF, and if it was associated with worse 30-day short-term outcomes. We also determined if age modified the association between delirium or cognitive impairment without delirium and 30-day outcomes.

## Material and methods

This was a pre-planned analysis of the Emergency Medicine Research Outcomes Consortium (EMROC) AHF registry, a prospective multicenter cohort study of subjects presenting with AHF to 6 EDs in the United States [16]. A convenience sample of patients was enrolled from September 2014 to March 2019. Patients were included in the EMOC AHF Registry within 12 hours of ED arrival if they were 18 years of age or older, English speaking, had a confirmed clinical diagnosis of AHF, and met any one of the following criteria: BNP >100 pg/mL, NT-proBNP > 900 pg/mL, radiographic or sonographic signs of pulmonary congestion, or treatment for AHF with intravenous diuretics or vasodilators. AHF diagnosis was adjudicated by a physician expert in heart failure after the ED visit using medical record review. For this analysis, only 4 out of 6 sites participated in the cognitive study (Vanderbilt University Medical Center, Detroit Medical Center–Detroit Receiving Hospital, Detroit Medical Center–Sinai Grace, and Indiana University–Eskenazi Hospital). The study protocol was approved by the institutional review boards of the respective enrolling centers, and all participating patients or their legally authorized representatives provided written informed consent.

### Primary outcome variable

Our primary outcome variable was a composite outcome consisting of 30-day all-cause death, 30-day all-cause rehospitalization, and prolonged index hospital length of stay. Components of the composite outcome were chosen because they are strong indicators of prognosis or resource utilization. Prolonged index hospital length of stay was incorporated because it is a

competing risk for 30-day all-cause rehospitalizations, i.e., patients who have prolonged hospitalizations are less likely to be rehospitalized within 30 days. The 7-day cut-point was the 75% percentile of hospital length of stay in our cohort. Secondary outcomes examined the individual components of the primary outcome, including 30-day all-cause mortality, 30-day all-cause rehospitalizations, and hospital length of stay. Outcomes were assessed by standardized phone follow-up or medical record review conducted by trained research staff.

## Cognitive impairment

Global cognition and delirium were ascertained at enrollment (in the ED) by trained research assistants. Training occurred via video didactics and practice cases (eddelirium.org). Global cognition was measured using the Short Blessed Test (SBT), which is a 6-item assessment assessing orientation, immediate and delayed memory, and attention; it is 95% sensitive and 65% for cognitive impairment [17]. Scores range from 0 to 28, with a score of 10 or more indicating the presence of cognitive impairment. Delirium was ascertained using the modified brief Confusion Assessment Method (bCAM) which consists of 4 features: (1) altered mental status and fluctuating course, (2) inattention, (3) altered level of consciousness, and 4) disorganized thinking. For a patient to meet criteria for delirium, both features 1 and 2, and either 3 or 4 must be present. The modified bCAM is 82% to 86% sensitive and 93% to 96% specific for delirium as diagnosed by a psychiatrist; the kappa between raters is 0.87, indicating excellent inter-observer reliability [18]. Patients were classified as delirious (bCAM positive regardless of SBT score), cognitively impaired without delirium (bCAM negative, SBT $\geq$ 10) or cognitively intact (bCAM negative and SBT $<$ 10). Because delirium causes an acute loss of cognition, the SBT would not reflect baseline cognition in delirious patients. Thus, pre-illness cognitive impairment could only be ascertained in patients without delirium.

## Additional variables collected

Education in years was collected prospectively. A past history of myocardial infarction hypertension, diabetes mellitus, chronic kidney disease, hemodialysis, dyslipidemia, and pulmonary hypertension was collected prospectively asking the patient or caregiver combined with medical record review. Acute HF mortality risk was estimated using age, heart rate, systolic blood pressure, and blood urea nitrogen using a risk score developed by Fonarow et al using the following formula: [19]

$$\begin{aligned} \text{log odds of mortality} \\ = (0.0212 * \text{blood urea nitrogen}) - (0.0192 * \text{systolic blood pressure}) \\ + (0.0131 * \text{heart rate}) + (0.0288 * \text{age}) - 4.72. \end{aligned}$$

Left ventricular ejection fraction (categorized as $<$40%) was also collected from transthoracic echocardiograms conducted during or within 12 months of index hospitalization. Missing ejection fraction was categorized as unknown.

## Data analysis

Measures of central tendency and dispersion for continuous variables were reported as medians and interquartile ranges. Categorical variables were reported as frequencies and proportions. Univariate comparisons between the three cognitive impairment groups were performed using the Kruskall-Wallis test for continuous or ordinal variables and the chi-squared test for categorical variables. To determine whether delirium or cognitive impairment without delirium was associated with the primary composite outcome of 30-day all-cause death, 30-day all-cause rehospitalization, or index hospital length of stay > 7 days,

multivariable logistic regression was performed. The primary independent variable as delirium and cognitive impairment without delirium with no cognitive impairment as the reference group. The model was adjusted for older age (dichotomized as ≥ 65 years old), black race, years of education, AHF mortality risk, past history of myocardial infarction, hypertension, diabetes mellitus, chronic kidney disease, dialysis, dyslipidemia, and pulmonary hypertension, and ejection fraction < 40%. These covariates were chosen based on expert opinion, literature review, and our previous work [6, 9]. We limited the number of covariates incorporated in the multivariable model to avoid overfitting [20]. Older age*delirium and older age*cognitive impairment without delirium interactions were also incorporated to the model to determine if being older or younger age modified the association between cognitive impairment and the primary outcome. To achieve the most parsimonious model, an interaction term was removed if the p-value was > 0.20. As a secondary analysis, we performed multivariable logistic regression for each component of the composite outcome. Models for 30-day all-cause rehospitalization and hospitalization > 7 day included the same covariates as the primary model. Because there were only 33 deaths at 30 days, the model for 30-day mortality was adjusted for AHF mortality risk and ejection fraction < 40% to avoid overfitting. Goodness-of-fit was assessed using the Pearson Chi-Square test for all models. Odds ratios (OR) with their 95% confidence intervals (95%CI) were reported. All statistical analyses were performed with SAS Enterprise Guide version 7.15 (SAS Institute, Carey, NC).

## Results

Of the 20,883 patients who presented with AHF in the participating EDs, 1500 patients participated in the EMROC Registry (**Fig 1**), and 1208 patients had an adjudicated diagnosis of AHF. Of these, 47 patients were excluded because they had had missing SBT or bCAM assessments, 38 patients were excluded because of missing covariate data, and 79 were excluded because they were lost to follow-up. Patients who had missing cognitive and baseline covariate data were more likely to be White and admitted to the intensive care unit (**S1 Table**). Of the 1044 patients included in this analysis, 87 (8.3%) were delirious, 254 (24.3%) had cognitive impairment without delirium, and 703 (67.3%) were cognitively intact at baseline. For the entire cohort, the median (IQR) age was 61 (52, 71) years old, 652 (62.5%) were Black, 462 (44.3%) were female, and 76 (7.3%) were discharged home.

Patient characteristics stratified by cognitive impairment status are provided in **Table 1**. In general, those with delirium or cognitive impairment without delirium were older and were more likely to be male or Black race than those who were cognitively intact. Patients with delirium were more likely to have an ejection fraction of < 40% and more likely to have a past history of myocardial infarction compared with patients with cognitive impairment without delirium and patients who were cognitively intact. There was no detectable difference in years of education among the three groups. Notably, the median (IQR) age for the delirium group was 64 (54, 71) years old. Of the 617 AHF patients who were < 65 years old, 47 (7.6%) met criteria for delirium and of the 427 AHF patients who were ≥ 65 years old, 40 (9.4%) met criteria for delirium. Similarly, the median age (IQR) of the cognitive impairment without delirium group was 65 (55, 76) years old; 126 (20.4%) of AHF patients < 65 years old and 128 (30.0%) ≥ 65 years old met criteria for cognitive impairment without delirium.

Patient characteristics stratified by 30-day composite outcome status can be seen in **S2 Table**. Patients who had the 30-day composite outcome were older, had higher SBT scores and HF mortality risk, were more likely to be on dialysis prior to enrollment, be admitted to the hospital, and be admitted to the ICU. The distribution of the proportion of the composite outcome and its components stratified by cognitive status are presented in **Fig 2.** The composite

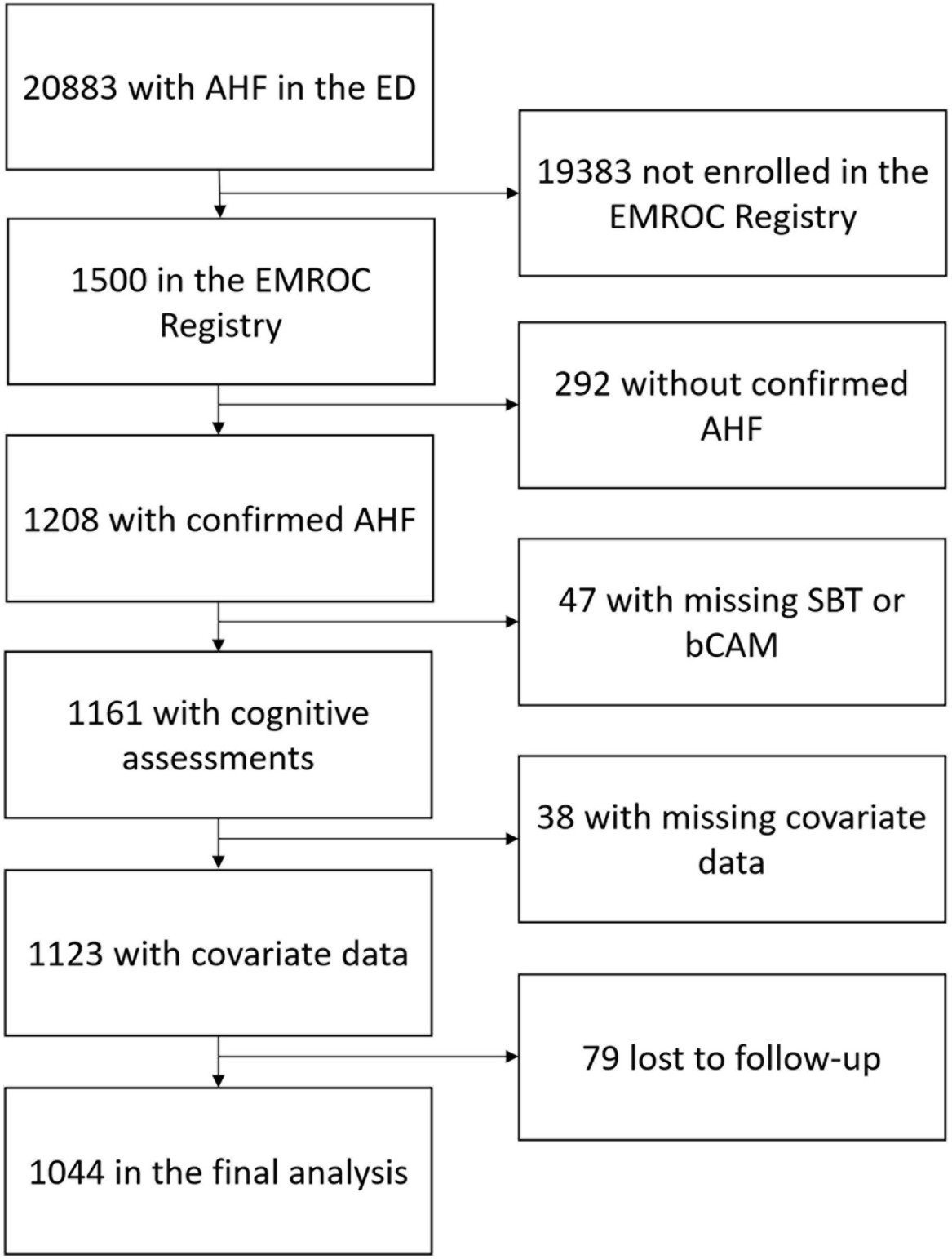

**Fig 1. Enrollment flow diagram.** AHF, acute heart failure; EMROC, Emergency Medicine Research Outcomes Consortium. SBT, Short Blessed Test; bCAM; Brief Confusion Assessment Method.

**Table 1. Patient characteristics stratified by cognitive impairment status at enrollment.** Cognitive impairment was determined by the Short Blessed Test (SBT); a score of 10 or more indicating the presence of cognitive impairment. Heart failure (HF) mortality risk was estimated using age, heart rate, systolic blood pressure, and blood urea nitrogen. Delirium was determined using the brief Confusion Assessment Method (bCAM). IQR, interquartile range; SBP, systolic blood pressure; BUN, blood urea nitrogen; ED, emergency department; ICU, intensive care unit.

| Variable | No Cognitive Impairment | Delirium | Cognitive Impairment Without Delirium N = 254 | P-value |
|---|---|---|---|---|
| | N = 703 | N = 87 | | |
| Median (IQR) Age, years | 60 (51, 69) | 64 (54, 71) | 65 (55, 76) | <0.0001 |
| Female, n (%) | 329 (46.8%) | 36 (41.4%) | 97 (38.2%) | 0.0517 |
| White | 277 (39.4%) | 23 (26.4%) | 66 (26.0%) | 0.0502 |
| Non-White Race, n (%) | 426 (60.6%) | 64 (73.6%) | 188 (74.0%) | |
| American Indian | 5 (0.7%) | 0 (0.0%) | 2 (0.8%) | |
| Asian | 3 (0.4%) | 0 (0.0%) | 1 (0.4%) | |
| Black | 406 (57.8%) | 64 (73.6%) | 182 (71.7%) | |
| Pacific Islander | 3 (0.4%) | 0 (0.0%) | 0 (0.0%) | |
| Other | 7 (1.0%) | 0 (0.0%) | 3 (1.2%) | |
| Unknown | 2 (0.3%) | 0 (0.0%) | 0 (0.0%) | |
| Median (IQR) Education, years | 13 (12, 14) | 12 (11, 13) | 12 (11, 13) | <0.0001 |
| Median (IQR) Short Blessed Test | 3 (2, 6) | 14 (11, 18) | 12 (10, 14) | <0.0001 |
| Median (IQR) HF Mortality Risk | 0.02 (0.01, 0.03) | 0.02 (0.01, 0.03) | 0.02 (0.01, 0.04) | 0.0010 |
| Median (IQR) SBP, mmHg | 148 (127, 174) | 144 (127, 157) | 148.5 (126, 169) | 0.2324 |
| Median (IQR) Heart Rate, beats per min | 91 (78, 104) | 89 (76, 105) | 88 (73, 101) | 0.0401 |
| Median (IQR) BUN, mg/dL | 21 (15, 31) | 24 (17, 34) | 23 (17, 34) | 0.0257 |
| Ejection Fraction < 40%, n (%) | 244 (34.7%) | 41 (47.1%) | 95 (37.4%) | 0.0705 |
| Past history, n (%) | | | | |
| Myocardial infarction | 190 (27.0%) | 36 (41.4%) | 74 (29.1%) | 0.0201 |
| Hypertension | 603 (85.8%) | 81 (93.1%) | 233 (91.7%) | 0.0132 |
| Diabetes Mellitus | 319 (45.4%) | 38 (43.7%) | 120 (47.2%) | 0.8118 |
| Dyslipidemia | 351 (49.9%) | 51 (58.6%) | 135 (53.2%) | 0.2547 |
| Chronic Kidney Disease | 230 (32.7%) | 32 (36.8%) | 97 (38.2%) | 0.2570 |
| Dialysis Dependent | 38 (5.4%) | 5 (5.8%) | 12 (4.7%) | 0.8972 |
| Pulmonary Hypertension | 65 (9.3%) | 6 (6.9%) | 13 (5.1%) | 0.1072 |
| Discharged Home from ED | 55 (7.8%) | 4 (4.6%) | 17 (6.7%) | 0.5054 |
| Ever Admitted to an ICU | 32 (12.6%) | 13 (14.9%) | 96 (13.7%) | 0.3617 |

outcome (death at 30-days, rehospitalization at 30-days, or index hospital length of stay > 7 days) was more frequent among patients with delirium occurring in 48 (55.2%) patients compared to 104 (40.9%) in the cognitively impaired without delirium group and 294 (41.8%) in the cognitively intact group. The proportion of delirious patients who experienced 30-day all-cause death and prolonged index hospitalization was higher compared to patients in the other two groups. The proportion of patients who were rehospitalized within 30 days was similar across cognitive function categories.

The **Fig 3** shows the adjusted odds ratios for delirium and cognitive impairment without delirium on the primary composite and secondary component outcomes; the reference group was cognitively intact patients. The odds ratios for the multivariable logistic regression model for the primary outcome can be seen in the **S3 Table**; the older age*delirium and older age*cognitive impairment without delirium interaction term p-values were 0.47 and 0.31, respectively. Therefore, these interaction terms were removed from the multivariable logistic regression model to maintain parsimony. Delirium, but not cognitive impairment without

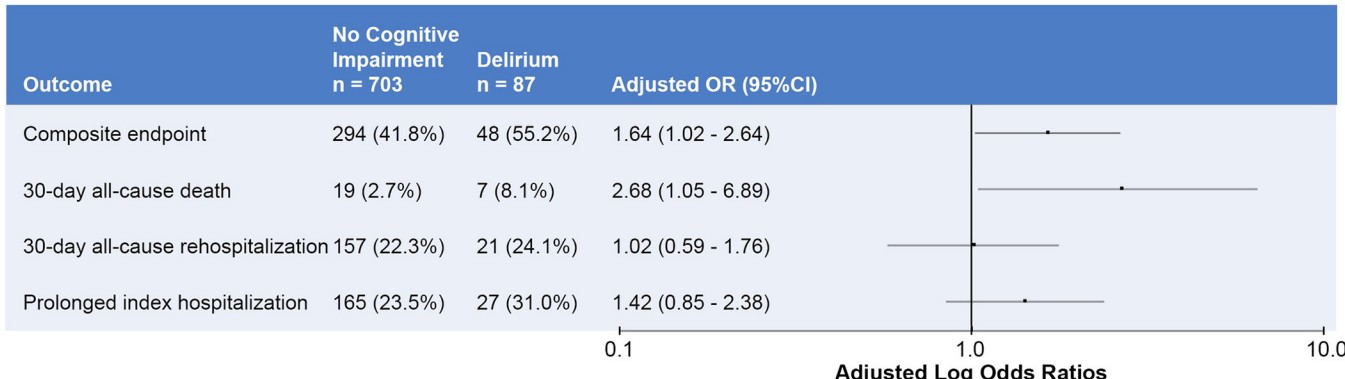

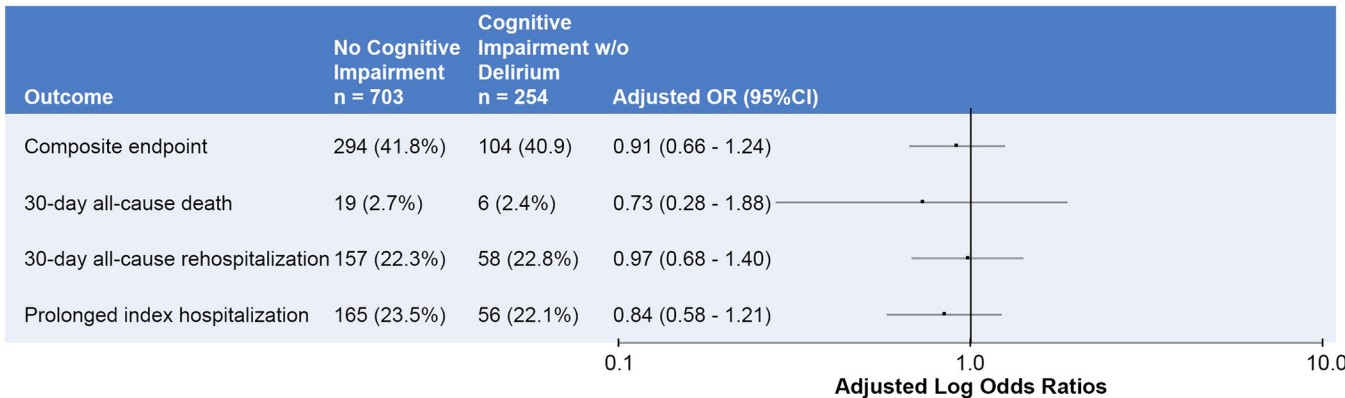

**Fig 2. Proportion of patient outcomes within each cognitive impairment category.** *p-value < 0.05. #, P-value < 0.10.

delirium, was significantly associated with increased odds of short-term adverse events (adjusted OR = 1.64, 95%CI: 1.02 to 2.64).

With regard to the secondary outcomes, patients with delirium had increased odds of 30-day all-cause mortality (adjusted OR = 2.68, 95%CI: 1.05–6.50. There was a trend towards increased odds of prolonged index hospitalizations in the delirium group, but this association was not statistically significant (adjusted OR = 1.42, 95% CI 0.85 to 2.39). We did not observe significant associations between delirium and 30-day rehospitalization. Cognitive impairment without delirium was not significantly associated with any of the secondary outcome components. Pearson Chi-Square p-values ranged from 0.2691 to 0.7338 for all multivariable logistic regression models indicating goodness-of-fit.

## Discussion

In our diverse cohort of 1044 ED patients with AHF, cognitive impairment secondary to delirium was an independent predictor of short-term adverse events, while cognitive impairment in the absence of delirium was not. These findings appear to be driven by the association between delirium and 30-day mortality and prolonged hospitalizations. Our findings suggest that delirium in patients with AHF is a marker of poor prognosis and increased short-term healthcare resource utilization. The median (IQR) age of 64 (54, 71) years old observed in the delirium cohort suggesting more than 50% of AHF patients with delirium were 64 years old or younger and 25% were 54 years old or younger; these are patients who are traditionally excluded from delirium studies. Older age, as defined as ≥ 65 years old, did not modify the

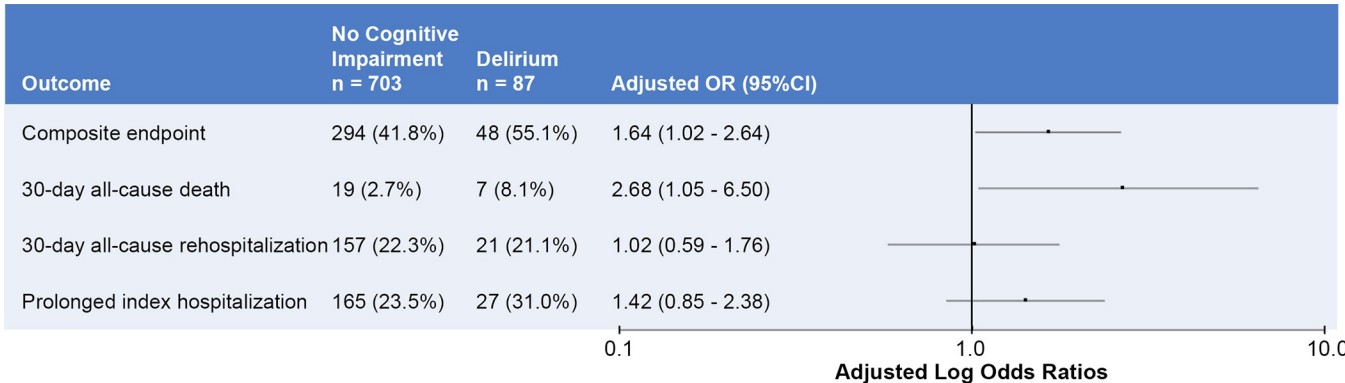

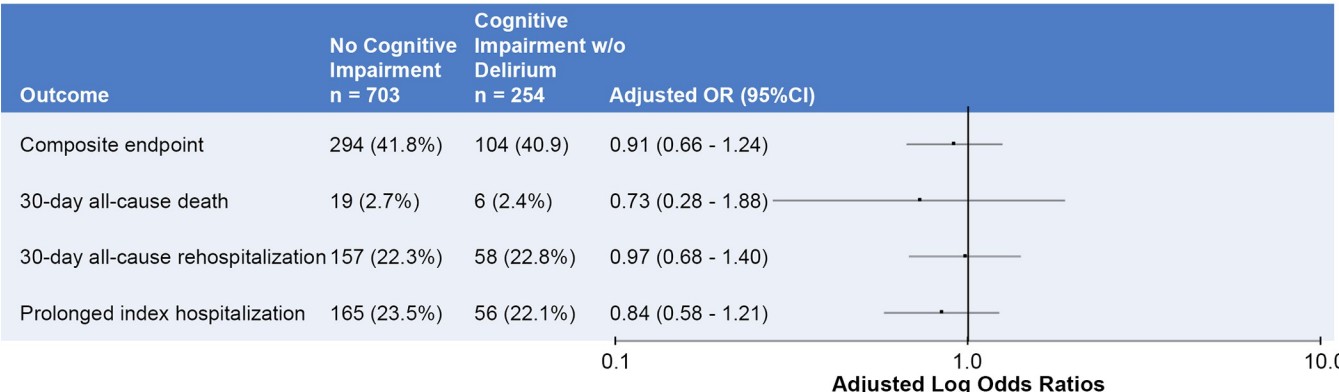

**Fig 3. Adjusted odds ratios for delirium and cognitive impairment without delirium on the primary and secondary outcomes.** The top panel represents delirium and the bottom panel represents cognitive impairment without delirium. OR, odds ratios; 95%CI, 95% confidence intervals.

association between delirium and adverse short-term outcomes indicating that delirium also portends poor prognosis in younger patients with AHF. Consequently, delirium should be considered in all patients with AHF regardless of age in the clinical setting and future investigations.

Several studies have previously reported that cognitive impairment in hospitalized patients with AHF is associated with death and hospital readmission [1, 21, 22]. However, cognitive impairment is a broad term that encompasses both acute (delirium) and chronic (dementia) cognitive deficits, which in the context of an acute illness, may have different therapeutic and prognostic significance. Previous studies similarly observed that delirium was associated with increased risk of rehospitalization, nursing home placement, and death in hospitalized AHF patients [11–15]. However, most were single-center studies that enrolled older patients (average age range: 75 to 83 years old) who were predominantly White or Asian race. They also did not include AHF patients who were discharged home. Our study extends the generalizability of their findings. We enrolled a more heterogeneous cohort of AHF patients across the entire adult age spectrum (median age of 61 years) and severity of illness from 4 EDs.

Additional studies are needed to understand delirium's full impact on AHF outcomes. Delirium has been shown to be an independent predictor of poorer long-term cognition in across different clinical settings [8, 23–25], but its impact on long-term cognitive trajectories in patients with acute AHF specifically remains unknown. If delirium negatively impacts cognitive health, additional consequences such as decreased health literacy and numeracy and increased risk of non-adherence to HF medications may exist. This confluence of events could

lead to a vicious cycle of additional AHF episodes, delirium, and hospitalizations, and further acceleration of cognitive decline. Importantly, this may also identify opportunities to intervene and prevent further cognitive decline. A better understanding of the inter-relationships between delirium, cognition, and chronic disease self-management may improve outcomes of patients with HF.

Surprisingly, younger age was not protective as the proportion of AHF patients with delirium were similar between older and younger patients. Age also did not modify the association between delirium and the 30-day composite outcome. Younger patients were more likely to have an ejection fraction < 40% (**S4 Table**) which can increase the patient's vulnerability to developing delirium due to brain hypoperfusion [26, 27]. Reduced ejection fraction also is a powerful predictor of all-cause mortality and may potentiate delirium's effect on long-term outcomes [28]. Younger patients with AHF were also more likely to self-identify as Black who are at higher risk for mortality and morbidity, and higher risk for heart failure-related hospitalizations compared with their non-Black counterparts [29, 30]. This is likely driven by health disparities and negative social determinants of health. Black patients with heart failure are more likely to have poorer access to health care and have a decreased likelihood of receiving guideline-recommended heart failure therapies [31, 32]. Black race is also associated with poorer heart failure disease self-management and decreased medication adherence [33], which can lead to AHF and subsequent hospitalizations. They are also more likely to have comorbidities such as diabetes, hypertension, and obesity which may further complicate disease self-management and medication adherence [34]. The complex interplay among age, health disparities, and social determinants of health may explain our findings and would have important implications for potential interventions aimed at improving HF outcomes in diverse patient populations.

Our study had many strengths. First, this was a multi-center cohort that was racially diverse with the majority being non-White. Second, we enrolled patients from the ED who represent a wide range of illness severity. Third, we used a highly specific method to identify patients with delirium. Strengths notwithstanding, our study had several limitations. We were limited by the low number of deaths and chose to use a composite outcome. Future studies with larger sample sizes are required to confirm these findings. The bCAM is 82% to 86% sensitive, and the SBT is 65% specific; misclassification may have occurred. This could potentially overestimate or attenuate the effect sizes. Delirium was assessed for at a single point in time. Because delirium can fluctuate, this may have underestimated the proportion of delirious patients. However, we feel the strength of estimating delirium upon hospital presentation is a unique aspect of our analysis and adds to the delirium literature regarding patients who are first assessed for delirium in the hospital. This could be part of a two pronged-approach, identifying and managing delirium in the ED and then following those at risk for development of delirium who screen negative in the ED. While we adjusted for several confounders for delirium and the adverse outcomes, residual confounding, e.g., due to co-infection or differences in polypharmacy, may still exist. Pre-illness cognition was not characterized in patients with delirium because this syndrome causes an acute loss of cognition. We did not use informant measures such as the short form Informant Questionnaire on Cognitive Decline in the Elderly score (IQCODE) [35]. Even in patients without delirium, the SBT may not have accurately reflected pre-illness cognition. Some of these patients may have had subsyndromal delirium or their scores were falsely low due to anxiety or multiple distractions (i.e., monitor alarms) commonly seen in the ED environment. While we adjusted for a past history of myocardial infarction, hypertension, diabetes mellitus, chronic kidney disease, dialysis, dyslipidemia, pulmonary hypertension, we did not capture all the comorbidities necessary to calculate the Charlson or Elixhauser comorbidity burden. We also did not account for pre-illness frailty or disability

which may have also led to additional residual confounding. The use of a convenience sample may have introduced selection bias. Additional selection bias may have been introduced because patient who had missing cognitive and baseline covariate data were more likely to be admitted to the intensive care unit.

## Conclusions

In conclusion, delirium and cognitive impairment without delirium were common in both younger and older patients with AHF. However, only delirium was significantly associated with the composite outcome for short-term adverse events consisting of 30-day mortality, 30-day rehospitalization, and prolonged index hospital length of stay. This association was not modified by age and was independent of traditional HF risk factors. Future studies should elucidate the mechanisms, such as poor medication adherence, responsible for this and develop interventions to improve outcomes in this vulnerable cohort.

## Supporting information

**S1 Table. Patient characteristics of patients who were in the analysis cohort, had missing cognition and other covariate data, and were lost to follow-up.** IQR, interquartile range; SBP, systolic blood pressure; BUN, blood urea nitrogen; ED, emergency department; ICU, intensive care unit.
(DOCX)

**S2 Table. Patient characteristics stratified by 30-day composite outcome status.** Heart failure (HF) mortality risk was estimated using age, heart rate, systolic blood pressure, and blood urea nitrogen. Delirium was determined using the brief Confusion Assessment Method (bCAM). IQR, interquartile range; SBP, systolic blood pressure; BUN, blood urea nitrogen; ED, emergency department; ICU, intensive care unit.
(DOCX)

**S3 Table. Multivariable logistic regression model for the primary composite outcome of 30-day all-cause death, 30-day all-cause rehospitalization, or hospitalization > 7 days.** Initially, older age*delirium and older age*cognitive impairment without delirium interaction terms were incorporated in the multivariable logistic regression model. Because, their p-values were 0.47 and 0.31, respectively, these interaction terms were removed from the model to maintain parsimony.
(DOCX)

**S4 Table. Patient characteristics stratified by older (≥ 65 years) versus younger (<65 years) age at enrollment.** Heart failure (HF) mortality risk was estimated using age, heart rate, systolic blood pressure, and blood urea nitrogen. Delirium was determined using the brief Confusion Assessment Method (bCAM). IQR, interquartile range; SBP, systolic blood pressure; BUN, blood urea nitrogen; ED, emergency department; ICU, intensive care unit.
(DOCX)

**S1 Data.**
(CSV)

## Acknowledgments

The Emergency Medicine Research and Outcomes Consortium (EMROC) Investigators consist of Drs. Sean Collins (Vanderbilt University Medical Center); Peter Pang (Indiana

University and Eskenazi Hospitals), Phillip Levy (Detroit Receiving and Sinai Grace Hospitals); and Gregory Fermann (University of Cincinnati Medical Center). Dr. Sean Collins (e-mail: sean.collins@vumc.org) leads the EMROC Investigators.

## Author Contributions

**Conceptualization:** Jin H. Han, Candace D. McNaughton, Peter S. Pang, Phillip D. Levy, Sean P. Collins.

**Data curation:** Jin H. Han, Peter S. Pang, Phillip D. Levy, Sarah Meram, Mette Lind Cole, Hadassah H. Paz, Kelly M. Moser.

**Formal analysis:** Jin H. Han, Cathy A. Jenkins.

**Funding acquisition:** Phillip D. Levy, Sean P. Collins.

**Investigation:** Jin H. Han.

**Project administration:** Karen F. Miller.

**Supervision:** Peter S. Pang, Phillip D. Levy, Karen F. Miller, Alan B. Storrow, Sean P. Collins.

**Writing – original draft:** Jin H. Han, Candace D. McNaughton, William B. Stubblefield, Sean P. Collins.

**Writing – review & editing:** Jin H. Han, Candace D. McNaughton, William B. Stubblefield, Peter S. Pang, Phillip D. Levy, Karen F. Miller, Sarah Meram, Mette Lind Cole, Cathy A. Jenkins, Hadassah H. Paz, Kelly M. Moser, Alan B. Storrow, Sean P. Collins.

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
