## [Decision Letter · Decision Letter 0]

22 Apr 2022

PONE-D-21-37051Delirium and Its Association with Short-term Outcomes in Younger and Older Patients with Acute Heart FailurePLOS ONE

Dear Dr. Han,

Thank you for submitting your manuscript to PLOS ONE. After careful consideration, we feel that it has merit but does not fully meet PLOS ONE’s publication criteria as it currently stands. Therefore, we invite you to submit a revised version of the manuscript that addresses the points raised during the review process.

ACADEMIC EDITOR: 

Among points raised by reviewers the following should be especially addressed:

Please describe better patient selection (“convenience sample” is not sufficient) and give an estimate of exclusion rateDescribe better among-group differences of HF mortality risk (the highly significant difference are not apparent from data in table 1)Include a supplementary analysis regarding the difference between older and younger patients and briefly discuss the reason of the lack of prognostic meaning of age and the similar prevalence of delirium between groups, both unexpected

Among limitations, you should include the lack of information regarding functional status (frailty and/or disability). Please consider to compute a comorbidity index to adjust for in a sensitivity analysis.

Moreover “cognitive impairment” does not represent premorbid cognitive impairment, also in subjects without delirium, and likely represents a mix between premorbid cognitive impairment, subsyndromal delirium and “false positives” (low education, anxiety, etc). Please discuss.

We look forward to receiving your revised manuscript.

Kind regards,

Enrico Mossello

Academic Editor

PLOS ONE

“This project was partially supported by the NCATS/NIH under award number UL1 TR000445.  Drs. Han and Collins receive funding from the Geriatric, Research, Education, and Clinical Center (GRECC).”

4. Thank you for stating the following in the Funding Section of your manuscript:

“This work was supported by the National Center for Research Resources, Grant UL1 RR024975-01, and is now at the National Center for Advancing Translational Sciences, Grant 2 UL1 TR000445-06. Dr. Han was also supported by the National Institutes of Health (NIH) under award number R21AG06312, R56HL141567 and R01AG065249. Drs. Han and McNaughton are also supported by the Veteran Affairs Geriatric Research, Education, and Clinical Center (GRECC). The content is solely the responsibility of the authors and does not necessarily represent the official views of Vanderbilt University Medical Center, NIH, and Veterans Affairs.”

“This project was partially supported by the NCATS/NIH under award number UL1 TR000445.  Drs. Han and Collins receive funding from the Geriatric, Research, Education, and Clinical Center (GRECC).”

5. One of the noted authors is a group or consortium [Emergency Medicine Research and Outcomes Consortium Investigators]. In addition to naming the author group, please list the individual authors and affiliations within this group in the acknowledgments section of your manuscript. Please also indicate clearly a lead author for this group along with a contact email address.

Reviewers' comments:

Reviewer's Responses to Questions

**Comments to the Author**

1. Is the manuscript technically sound, and do the data support the conclusions?

Reviewer #1: Yes

Reviewer #2: Yes

2. Has the statistical analysis been performed appropriately and rigorously? 

Reviewer #1: Yes

Reviewer #2: Yes

3. Have the authors made all data underlying the findings in their manuscript fully available?

Reviewer #1: Yes

Reviewer #2: Yes

4. Is the manuscript presented in an intelligible fashion and written in standard English?

Reviewer #1: Yes

Reviewer #2: Yes

5. Review Comments to the Author

Reviewer #1: Dr. Jin Ho Han and colleagues should be congratulated for addressing the important issue of early prognostic impact of incident delirium in patients admitted for AHF in older and younger patients. This represents a novelty indeed because delirium has rarely been assessed in younger subjects.

The authors identify three main groups: delirious (bCAM positive regardless of SBT score), cognitively impaired without delirium (bCAM negative, SBT > 10) or cognitive intact (bCAM negative and SBT < 10).

Nevertheless my main concerns are related to possible misclassification of Delirium/ cognitive impairment and possibly related to the way these condition were classified.

The rate of delirium seems similar to other literature reports while the presence of cognitive impairment seem very high (considering an overall quite young population) with one out four patients being classified with this condition. The authors may want to comment on this and on the quite unexpected similar incidence of delirium among younger and older patients.

Furthermore, I do not understand why the authors say that pre-illness cognitive impairment could be ascertained in patients without delirium. Authors should better clarify this point since a history of cognitive impairment (investigated also with family members and/or caregivers) might have increased accuracy in classifications.

Other minor considerations

Global cognition was measured using the Short Blessed Test (SBT), which is a 6-item assessment assessing orientation, immediate and delayed memory, and attention; it is 95% sensitive and 65% for cognitive impairment [17]. Scores range from 0 to 28, with a score of 10 or more indicating the presence of cognitive impairment.

add specific after 65%

Prolonged index hospital length of stay was incorporated because it is a competing risk for 30-day all-cause rehospitalizations, i.e., patients who have 5 prolonged hospitalizations are less likely to be rehospitalized within 30 days.

This is not true… Data from the literature show that there is a U wave relation between hospitalization length and early readmissions: short length of stay correlates with 30-days HF-hospitalizations, while long length of stay is correlated with risk

Reviewer #2: Thank you for inviting me to review this study that investigated the association of delirium with short-term adverse outcomes in patients with HF. The study investigated an important topic involving cognitive impairment that is highly prevalent and affects prognosis in patients with HF. I have a few comments as follow.

1. Were consecutive patients recruited into the study?

2. Prolonged hospital stay is likely due to greater severity of HF that may be also the cause of acute delirium. So including prolonged hospital stay may have overestimated the association of delirium with short-term outcomes in your study. Prolonged hospital stay might be associated with lower readmission because of more complete treatment of HF and management of comorbidity. Patients may be discharged with optimal fluid status. Prolonged stay may be a predictor of readmission, but not a competing risk.

3. How did the authors ensure that readmissions to other hospitals were not missed?

4. Why did the authors adjust for age as a binary variable, but not as a continuous variable? Did the authors also check for interaction with continuous age variable?

5. Did the delirium resolve when patients were discharged from hospital?

6. Delirium was only significantly associated with all-cause death, which reflects a greater level of HF severity and frailty in this group of patients. The HF mortality score presented in this study was not very informative because they looked largely similar across 3 groups of patients but had a highly significant p-value. What was the comorbidity index for your patients?

6. PLOS authors have the option to publish the peer review history of their article (what does this mean?). If published, this will include your full peer review and any attached files.

Reviewer #1: No

Reviewer #2: No

---

## [Author Response · Author response to Decision Letter 0]

4 Jun 2022

Thank you for the detailed reviews and for allowing us to revise and resubmit this manuscript. We have addressed all the suggestions/criticisms below and we hope that you find our revisions satisfactory. The details of our changes are provided in the subsequent pages; our responses are preceded by “***”. Thank you again for allowing us the opportunity to revise and resubmit this manuscript. If you have any questions, please don’t hesitate to contact me by phone (615-260-0086) or by e-mail. 

Please describe better patient selection ("convenience sample" is not sufficient) and give an estimate of exclusion rate.

***Thank you for this suggestion. We added an Enrollment Flow Diagram (Figure 1) to better characterize our exclusions. We also added the following (see underlined) to the Results on Page 7:

“Of the 20,883 patients who presented with AHF in the participating EDs, 1500 patients participated in the EMROC Registry (Figure 1), and 1208 patients had an adjudicated diagnosis of AHF. Of these, 47 patients were excluded because they had had missing SBT or bCAM assessments, 38 patients were excluded because of missing covariate data, and 79 were excluded because they were lost to follow-up.”

Describe better among-group differences of HF mortality risk (the highly significant difference are not apparent from data in table 1).

***Per your suggestion, we have added Supplemental Table 2 which stratifies patient characteristics by 30-day composite outcome status. In the Results section (Page 10), we also added the following text:

“Patient characteristics stratified by 30-day composite outcome status can be seen in Supplementary Table 2. Patients who had the 30-day composite outcome were older, had higher SBT scores and mortality risk, and were more likely to be on dialysis prior to enrollment, be admitted to the hospital, and be admitted to the ICU.”

Include a supplementary analysis regarding the difference between older and younger patients and briefly discuss the reason of the lack of prognostic meaning of age and the similar prevalence of delirium between groups, both unexpected 

***Thank you for this suggestion. We have added Supplemental Table 4. Patient characteristics stratified by older (> 65 years) versus younger (<65 years) at enrollment. We also added the following paragraph to the discussion section to provided potential reasons:

“Surprisingly, younger age was not protective as the proportion of AHF patients with delirium were similar between older and younger patients. Age also did not modify the association between delirium and the 30-day composite outcome. Younger patients were more likely to have an ejection fraction < 40% (Supplementary Table 4) which can increase the patient’s vulnerability to developing delirium due to brain hypoperfusion [26, 27]. Reduced ejection fraction also is a powerful predictor of all-cause mortality and may potentiate delirium’s effect on long-term outcomes [28]. Younger patients with AHF were also more likely to self-identify as Black who are at higher risk for mortality and morbidity, and higher risk for heart failure-related hospitalizations compared with their non-Black counterparts [29, 30]. This is likely driven by health disparities and negative social determinants of health. Black patients with jeart failure are more likely to have poorer access to health care and have a decreased likelihood of receiving guideline-recommended heart failure therapies [31, 32]. Black race is also associated with poorer heart failure disease self-management and decreased medication adherence [33], which can lead to AHF and subsequent hospitalizations. They are also more likely to have comorbidities such as diabetes, hypertension, and obesity which may further complicate disease self-management and medication adherence [34]. The complex interplay among age, health disparities, and social determinants of health may explain our findings and would have important implications for potential interventions aimed at improving HF outcomes in diverse patient populations.”

Among limitations, you should include the lack of information regarding functional status (frailty and/or disability). 

***Thank you for pointing this out. We added the following sentence to the limitations section:

“We also did not account for pre-illness comorbidity burden, frailty or disability which may have led to additional residual confounding.”

Please consider to compute a comorbidity index to adjust for in a sensitivity analysis.

***Our registry collected a past history of myocardial infarction, hypertension, diabetes mellitus, chronic kidney disease, dialysis, dyslipidemia, pulmonary hypertension, and ejection fraction < 40%. These were adjusted in the multivariable models since we had sufficient degrees of freedom. Unfortunately, we did not collect all the variables necessary to calculate the Charlson or Elixhauser comorbidity burden and this is a limitation. We added the following sentence to the limitations section:

“While we adjusted for a past history of myocardial infarction, hypertension, diabetes mellitus, chronic kidney disease, dialysis, dyslipidemia, pulmonary hypertension, we did not capture all the comorbidities necessary to calculate the Charlson or Elixhauser comorbidity burden. We also did not account for pre-illness frailty or disability. The lack of adjustment for these covariates may have led to additional residual confounding. 

Moreover "cognitive impairment" does not represent premorbid cognitive impairment, also in subjects without delirium, and likely represents a mix between premorbid cognitive impairment, subsyndromal delirium and "false positives" (low education, anxiety, etc). Please discuss.

***In the limitations, the following sentence were added:

“Even in patients without delirium, the SBT may not have accurately reflected pre-illness cognition. Some of these patients may have had subsyndromal delirium or their scores were falsely low due to anxiety or multiple distractions (i.e.,monitor alarms) commonly seen in the ED environment.”

Editorial Office Comments

1. Thank you for stating the following financial disclosure:

"This project was partially supported by the NCATS/NIH under award number UL1 TR000445. Drs. Han and Collins receive funding from the Geriatric, Research, Education, and Clinical Center (GRECC)."

*** The funders had no role in study design, data collection and analysis, decision to publish, or preparation of the manuscript. We have added this to the cover letter. Thank you for changing the online submission form on our Behalf.

2. One of the noted authors is a group or consortium [Emergency Medicine Research and Outcomes Consortium Investigators]. In addition to naming the author group, please list the individual authors and affiliations within this group in the acknowledgments section of your manuscript. Please also indicate clearly a lead author for this group along with a contact email address.

***We have added the following to the Acknowledgements:

“The Emergency Medicine Research and Outcomes Consortium (EMROC) Investigators consist of Drs. Sean Collins (Vanderbilt University Medical Center); Peter Pang (Indiana University and Eskenazi Hospitals), Phillip Levy (Detroit Receiving and Sinai Grace Hospitals); and Gregory Fermann (University of Cincinnati Medical Center). Dr. Sean Collins (e-mail: sean.collins@vumc.org) leads the EMROC Investigators.”

---

## [Editor Report · Decision Letter 1]

20 Jun 2022

Delirium and Its Association with Short-term Outcomes in Younger and Older Patients with Acute Heart Failure

PONE-D-21-37051R1

Dear Dr. Han,

We’re pleased to inform you that your manuscript has been judged scientifically suitable for publication and will be formally accepted for publication once it meets all outstanding technical requirements.

Kind regards,

Enrico Mossello

Academic Editor

PLOS ONE
---

## [Editor Report · Acceptance letter]

15 Jul 2022

PONE-D-21-37051R1 

Delirium and Its Association with Short-term Outcomes in Younger and Older Patients with Acute Heart Failure 

Dear Dr. Han:

I'm pleased to inform you that your manuscript has been deemed suitable for publication in PLOS ONE. Congratulations! Your manuscript is now with our production department. 

Kind regards, 

on behalf of

Dr. Enrico Mossello 

Academic Editor

PLOS ONE